# Excitatory transmission onto AgRP neurons is regulated by cJun NH$_2$-terminal kinase 3 in response to metabolic stress

**Santiago Vernia[1†‡], Caroline Morel[1†], Joseph C Madara[2,3], Julie Cavanagh-Kyros[1,4], Tamera Barrett[1,4], Kathryn Chase[5], Norman J Kennedy[1], Dae Young Jung[1], Jason K Kim[1,5], Neil Aronin[5], Richard A Flavell[6,7], Bradford B Lowell[2,3], Roger J Davis[1,4*]**

[1]Program in Molecular Medicine, University of Massachusetts Medical School, Worcester, United States; [2]Division of Endocrinology, Beth Israel Deaconess Medical Center, Boston, United States; [3]Harvard Medical School, Boston, United States; [4]Howard Hughes Medical Institute, University of Massachusetts Medical School, Worcester, United States; [5]Department of Medicine, Division of Endocrinology, University of Massachusetts Medical School, Worcester, United States; [6]Department of Immunobiology, Yale University School of Medicine, New Haven, United States; [7]Howard Hughes Medical Institute, Yale University School of Medicine, New Haven, United States

**\*For correspondence:** roger.davis@umassmed.edu

[†]These authors contributed equally to this work

**Present address:** [‡]Genes and Metabolism Section, MRC Clinical Sciences Centre, Imperial College London, Hammersmith Campus, London, United Kingdom

**Abstract** The cJun NH$_2$-terminal kinase (JNK) signaling pathway is implicated in the response to metabolic stress. Indeed, it is established that the ubiquitously expressed JNK1 and JNK2 isoforms regulate energy expenditure and insulin resistance. However, the role of the neuron-specific isoform JNK3 is unclear. Here we demonstrate that JNK3 deficiency causes hyperphagia selectively in high fat diet (HFD)-fed mice. JNK3 deficiency in neurons that express the leptin receptor LEPRb was sufficient to cause HFD-dependent hyperphagia. Studies of sub-groups of leptin-responsive neurons demonstrated that JNK3 deficiency in AgRP neurons, but not POMC neurons, was sufficient to cause the hyperphagic response. These effects of JNK3 deficiency were associated with enhanced excitatory signaling by AgRP neurons in HFD-fed mice. JNK3 therefore provides a mechanism that contributes to homeostatic regulation of energy balance in response to metabolic stress.

## Introduction

The regulation of energy balance (food consumption and energy expenditure) is important for health and survival. Sustained negative energy balance caused by cachexia and anorexia is associated with serious injury to multiple organ systems (*Aoyagi et al., 2015*; *Mehler and Brown, 2015*). Similarly, sustained positive energy balance caused by hyperphagia results in obesity associated with severe metabolic disorders (e.g. type 2 diabetes, cardiovascular disease, hepatitis, neurodegeneration and cancer) that represent leading causes of morbidity and mortality (*Flegal et al., 2013*). The homeo-static maintenance of energy balance is therefore critically important.

It is established that the arcuate nucleus (ARC) in the hypothalamus plays a key role in the regulation of energy balance (*Cone, 2005*). AgRP neurons in the ARC mediate orexigenic signals, including neuropeptide Y (NPY), agouti-related peptide (AgRP), and γ-aminobutyric acid (GABA) that project

**eLife digest** Consuming the right amount of food is important for health. Eating too little for a long time causes damage to organs, and overeating can cause harm as well, in the form of conditions such as obesity and type 2 diabetes. Several signaling molecules and brain regions are linked to controlling food consumption and ensuring the body receives the correct amount of nutrients to fuel its activities.

Previous studies have found that two proteins called JNK1 and JNK2, which are found in most tissues of the body, can reduce how much energy cells use. This can trigger insulin resistance and fat accumulation, and so suggests that blocking the activity of these proteins may help to treat type 2 diabetes and obesity. However, the role of another JNK protein – JNK3, which is mostly found in the brain – was not known.

Now, Vernia, Morel et al. have investigated the role of JNK3 in metabolism. It was found that JNK3 reduced the amount of food consumed by mice provided with a cafeteria (high fat) diet. Mice that lacked JNK3 ate far more food and gained more weight on a high fat diet than normal mice. However, JNK3 played no role in food consumption when mice were fed a standard chow diet. Treating normal mice with leptin – an appetite-suppressing hormone – caused them to lose weight, but did not affect mice that lacked JNK3.

Examining the brains of the mice revealed that in normal mice, JNK3 in a specific sub-population of neurons decreases the production of proteins that promote eating. However, the proteins continued to be produced in mice that lacked JNK3, encouraging overeating.

Overall, the results suggest that blocking the activity of all the JNK proteins will not help treat obesity and diabetes as shutting down JNK3 could encourage overeating. Therefore, future investigation into treatments for these conditions should focus on drugs that specifically target JNK1 and JNK2, and not JNK3.

to POMC neurons in the ARC and to secondary response neurons in many brain regions, including the lateral hypothalamus (LH) and the paraventricular nucleus (PVN) of the hypothalamus. In contrast, POMC neurons mediate anorexigenic signals, including cocaine and amphetamine regulated transcript (CART) and pro-opiomelanocortin (POMC)-derived α-melanocyte stimulating hormone (α-MSH). POMC neurons project to many brain areas, including the PVN and LH in the hypothalamus where α-MSH acts as an agonist of the melanocortin receptors MC3R and MC4R on secondary response neurons to inhibit feeding and increase energy expenditure. Importantly, this action of α-MSH is antagonized by AgRP. Moreover, POMC neurons receive inhibitory GABAergic input from AgRP neurons. Consequently, AgRP and POMC neurons act together to balance food consumption, energy expenditure and nutrient homeostasis (*Cone, 2005*).

AgRP and POMC neurons integrate signals from nutrients (e.g. glucose and fatty acids) and peripheral hormones (e.g. leptin, insulin, ghrelin, and cytokines) to mediate opposite actions regulating downstream neuroendocrine circuits linking internal and environmental stimuli with the coordinated control of homeostatic satiety (*Blouet and Schwartz, 2010*; *Varela and Horvath, 2012*). Thus, leptin activates POMC neurons (*Cowley et al., 2001*) and inhibits AgRP neurons (*Takahashi and Cone, 2005*) leading to reduced food consumption and increased energy expenditure. These processes can be regulated by intracellular signaling networks, including the Janus kinase 2-signal transducer and activator of transcription 3 (JAK2-STAT3) axis (*Bates and Myers, 2003*), Rho-associated coiled coil containing protein kinase 1 (ROCK1) (*Huang et al., 2012*), mechanistic target of rapamycin (mTOR) (*Mori et al., 2009*; *Kocalis et al., 2014*), adenosine monophosphate-activated protein kinase (AMPK) (*Claret et al., 2007*; *Dagon et al., 2012*), and phosphatidylinositol-4,5-bisphosphate 3-kinase (PI3K) (*Niswender et al., 2003*), that contribute to the fine-tuning of energy balance.

The anorexigenic hormone leptin plays a key role in the regulation of food consumption. Leptin can act directly on AgRP and POMC neurons, but leptin can also act on other neurons in several brain sub-regions, including mid-brain and brainstem nuclei (*Scott et al., 2009*; *Patterson et al., 2011*). Control of leptin signaling in these neurons is important for maintaining energy balance. For

example, obesity causes an increase in the blood concentration of leptin, most likely because of increased adipose tissue mass. The increased leptin concentration can lead to tachyphylaxis and suppression of the anorexigenic actions of leptin (*Frederich et al., 1995*). This mechanism enables homeostatic regulation of feeding behavior in response to metabolic stress. Whether this mechanism represents "leptin resistance" is unclear (*Myers et al., 2010*) because some biochemical aspects of leptin signaling are maintained in the obese state (*Ottaway et al., 2015*). A requirement for leptin signaling may reflect the role of the leptin-stimulated JAK2-STAT3 pathway to increase expression of the negative regulator SOCS3 (*Allison and Myers, 2014*). Negative regulation of leptin signaling may also involve the tyrosine phosphatases PTPN1 and PTPN2 (*Bence et al., 2006*; *Loh et al., 2011*), reactive oxygen species (*Diano et al., 2011*), the endoplasmic reticulum unfolded protein response (*Zhang et al., 2008*; *Ozcan et al., 2009*), autophagy (*Kaushik et al., 2011*), and low-grade inflammation (*de Git and Adan, 2015*).

The purpose of the study reported here was to test whether the cJun $NH_2$-terminal kinase (JNK) signaling pathway regulates feeding behavior. Previous studies have established that the ubiquitously expressed JNK1 and JNK2 isoforms play an important role in the metabolic stress response of peripheral tissues (*Sabio and Davis, 2010*). However, loss-of-function studies have not identified a role for JNK in the control of food consumption. Here we demonstrate that the neuronal isoform JNK3 (encoded by the *Mapk10* gene) plays a key role in the maintenance of energy balance during consumption of a high fat diet (HFD) by promoting leptin signaling. *Mapk10* gene ablation studies identify AgRP neurons as a site of JNK3 function. JNK3 is therefore a key mediator of homeostatic regulation of energy balance in response to metabolic stress.

## Results

### Feeding a high fat diet causes JNK3 activation

Leptin is an anorexigenic hormone. Indeed, treatment of chow-fed mice with leptin suppressed feeding behavior and caused decreased body mass (*Figure 1A*). In contrast, HFD-fed mice failed to respond to leptin (*Figure 1A*). The mechanism that accounts for this observation is unclear, but may involve both decreased leptin signaling and reduced signaling by down-stream mediators (e.g. MC4R). Tachyphylaxis may be a contributing factor and mutational analysis of leptin signaling components implicates functions of the leptin receptor, tyrosine phosphatases, reactive oxygen species, and SOCS3 (*Myers et al., 2010*).

We considered the possibility that a stress-activated MAP kinase pathway may contribute to the regulation of leptin signaling in HFD-fed mice. It is established that feeding a HFD causes activation of the ubiquitously expressed isoforms JNK1 and JNK2 in peripheral tissues, including liver, muscle, and adipose tissue (*Sabio and Davis, 2010*). However, the regulation of JNK caused by feeding a HFD in the central nervous system is unclear because these ubiquitously expressed JNK isoforms in neurons are constitutively activated and are primarily localized to axons and dendrites (*Coffey et al., 2000*; *Oliva et al., 2006*). In contrast, the neuron-specific isoform JNK3 exhibits low basal activity and can be activated in the nucleus when neurons are exposed to environmental stress (*Yang et al., 1997*). We therefore tested whether feeding a HFD caused activation of JNK3. This analysis demonstrated that feeding a HFD caused JNK3 phosphorylation and activation in the hypothalamus (*Figure 1B*). JNK3 in the central nervous system is therefore responsive to diet-induced metabolic stress. This JNK3 pathway represents a possible mediator of altered leptin signaling in HFD-fed mice.

### JNK3 deficiency promotes obesity and insulin resistance

To examine the role of the JNK3 pathway, we investigated the effect of feeding a chow diet (CD) or a HFD to wild-type (WT) mice or *Mapk10*[-/-] (JNK3-deficient) mice. We found that *Mapk10*[-/-] mice gained similar body mass when fed a CD, but these mice gained significantly greater mass when fed a HFD compared with WT mice (*Figure 1C*). [1]H-MRS analysis demonstrated that the greater HFD-induced body mass was caused by increased fat and lean mass (*Figure 1D*). Indeed, HFD-fed *Mapk10*[-/-] mice exhibited increased liver, skeletal muscle, heart, and adipose tissue mass compared with HFD-fed WT mice (*Figure 1—figure supplement 1A*). Microscopic examination of tissue

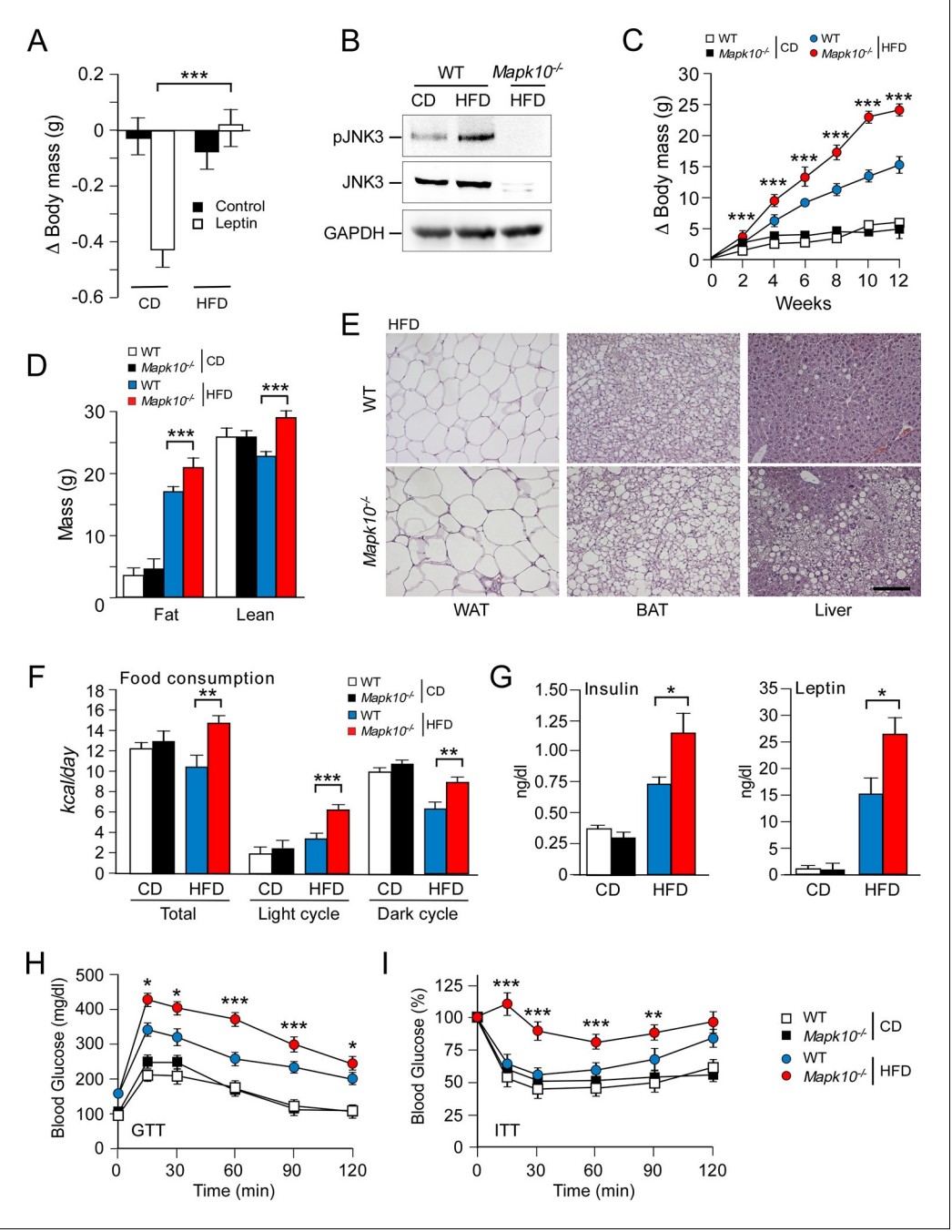

**Figure 1.** JNK3 deficiency causes hyperphagia and obesity. (**A**) WT mice were fed (4 wk) a chow diet (CD) or a high-fat diet (HFD). The body mass change at 24 hr post-injection (i.p. with solvent (PBS) or 2.5 mg/kg leptin) was measured (mean ± SEM; n=8; ***p<0.001). (**B**) WT and *Mapk10⁻/⁻* mice fed (12 wk) a CD or a HFD were starved overnight. Phospho-JNK3, JNK3, and GAPDH in the hypothalamus were measured by immunoblot analysis. (**C,D**) The body mass gain of CD-fed and HFD-fed (12 wk) mice was measured (mean ± SEM; n=10~12) (**C**). Fat and lean mass were measured by ¹H-MRS analysis (mean ± SEM; n=10~12). (**D**) Statistically significant differences between WT and *Mapk10⁻/⁻* mice are indicated (***p<0.001). (**E**) Paraffin embedded sections of epididymal white adipose tissue (WAT), interscapular brown adipose tissue (BAT), and liver were prepared from HFD-fed (12 wk) WT and *Mapk10⁻/⁻* mice. The sections were stained with hematoxylin & eosin. Scale bar, 100 µm. (**F**) Food consumption by WT and *Mapk10⁻/⁻* mice fed a CD or a HFD (3 wk) was measured (mean ± SEM; n=6; **p<0.01; ***p<0.001). (**G**) WT and *Mapk10⁻/⁻* mice fed a CD or a HFD (4 wk) were fasted overnight and the blood concentration of leptin and insulin was measured (mean ± SE; n=10~12; *p<0.05). (**H,I**) Glucose tolerance tests (**H**) and insulin tolerance tests (**I**)

*Figure 1 continued on next page*

*Figure 1 continued*

were performed on WT and *Mapk10*$^{-/-}$ mice fed a CD or a HFD (12 wk) by measurement of blood glucose concentration (mean ± SEM; n=10~12; *p < 0.05; **p < 0.01; ***p < 0.001).

The following figure supplements are available for figure 1:

**Figure supplement 1.** JNK3 deficiency causes obesity without changes in energy expenditure.
**Figure supplement 2.** Time course of the development of hyperphagia in HFD-fed JNK3-deficient mice.
**Figure supplement 3.** Increased food consumption is required for obesity caused by JNK3 deficiency in HFD-fed mice.

sections demonstrated increased hypertrophy of white and brown adipocytes and increased hepatic steatosis in HFD-fed *Mapk10*$^{-/-}$ mice compared with HFD-fed WT mice (*Figure 1E*).

We performed metabolic cage analysis to examine the mechanism of obesity promoted by JNK3 deficiency. These studies demonstrated that *Mapk10* gene ablation selectively increased consumption of a HFD, but not a CD (*Figure 1F*). Time course analysis demonstrated that the HFD-selective hyperphagia was observed within 2 days of consuming the HFD (*Figure 1—figure supplement 2A*) and was detected prior to the development of obesity (*Figure 1—figure supplement 2B*). No significant changes in V$_{O2}$, V$_{CO2}$, or energy expenditure were detected in the HFD-fed mice (*Figure 1—figure supplement 1B*). These data suggest that hyperphagia contributes to the increased obesity of HFD-fed *Mapk10*$^{-/-}$ mice compared with HFD-fed WT mice.

We used a pair-feeding protocol to test whether the increased obesity of *Mapk10*$^{-/-}$ mice compared with WT mice was caused by greater food consumption. We found that WT and *Mapk10*$^{-/-}$ mice gained similar body mass when fed the same amount of food (*Figure 1—figure supplement 3*). These data demonstrate that hyperphagia accounts for the increased HFD-induced obesity of *Mapk10*$^{-/-}$ mice compared with WT mice.

Consequences of the increased HFD feeding behavior of *Mapk10*$^{-/-}$ mice include increased hyperinsulinemia and hyperleptinemia (*Figure 1G*), increased blood lipid concentrations (*Figure 1—figure supplement 1C*), decreased glucose tolerance (*Figure 1H*), and increased insulin resistance (*Figure 1I*) when fed a HFD, but not a CD. These data indicate that *Mapk10*$^{-/-}$ mice may exhibit increased HFD-induced insulin resistance. To test this hypothesis, we performed a hyperinsulinemic-euglycemic clamp study. No significant differences between CD-fed WT and *Mapk10*$^{-/-}$ mice were detected (*Figure 2A–F*). In contrast, HFD-fed *Mapk10*$^{-/-}$ mice showed significantly reduced glucose infusion rate (a measure of whole body insulin sensitivity), reduced glucose turnover, reduced whole body glycolysis, increased hepatic glucose production, and decreased hepatic insulin action compared with HFD-fed WT mice (*Figure 2A–F*). These data demonstrate that *Mapk10*$^{-/-}$ mice exhibit a profound defect in glycemic regulation compared with WT mice when fed a HFD, but not a CD.

## JNK3 deficiency promotes adipose tissue inflammation

The increased adipose tissue mass of HFD-fed *Mapk10*$^{-/-}$ mice compared with HFD-fed control mice was associated with increased adipose tissue infiltration by F4/80$^+$ macrophages (*Figure 2G*). Indeed, gene expression analysis identified markedly increased expression of macrophage marker genes (*Emr1* (F4/80) & *Cd68*), increased expression of genes associated with M1-like macrophage polarization (*Ccl2, Il1b, Il6* & *Tnf*), and decreased expression of genes associated with M2-like macrophage polarization (*Arg1, Mgl2, Mrc1* & *Mrc2*) in the adipose tissue of HFD-fed *Mapk10*$^{-/-}$ mice compared with HFD-fed control mice (*Figure 2H–J*). These data indicate that JNK3 deficiency promotes increased adipose tissue inflammation in HFD-fed mice. It is likely that this increase in inflammation contributes to the glucose intolerant and insulin resistant phenotype of HFD-fed *Mapk10*$^{-/-}$ mice compared with HFD-fed WT mice (*Brestoff and Artis, 2015*).

## JNK3 deficiency suppresses leptin signaling

Low concentrations of leptin were detected in the blood when WT and *Mapk10*$^{-/-}$ mice were fed a CD (*Figure 1G*). The blood leptin concentration was increased when these mice were fed a HFD and

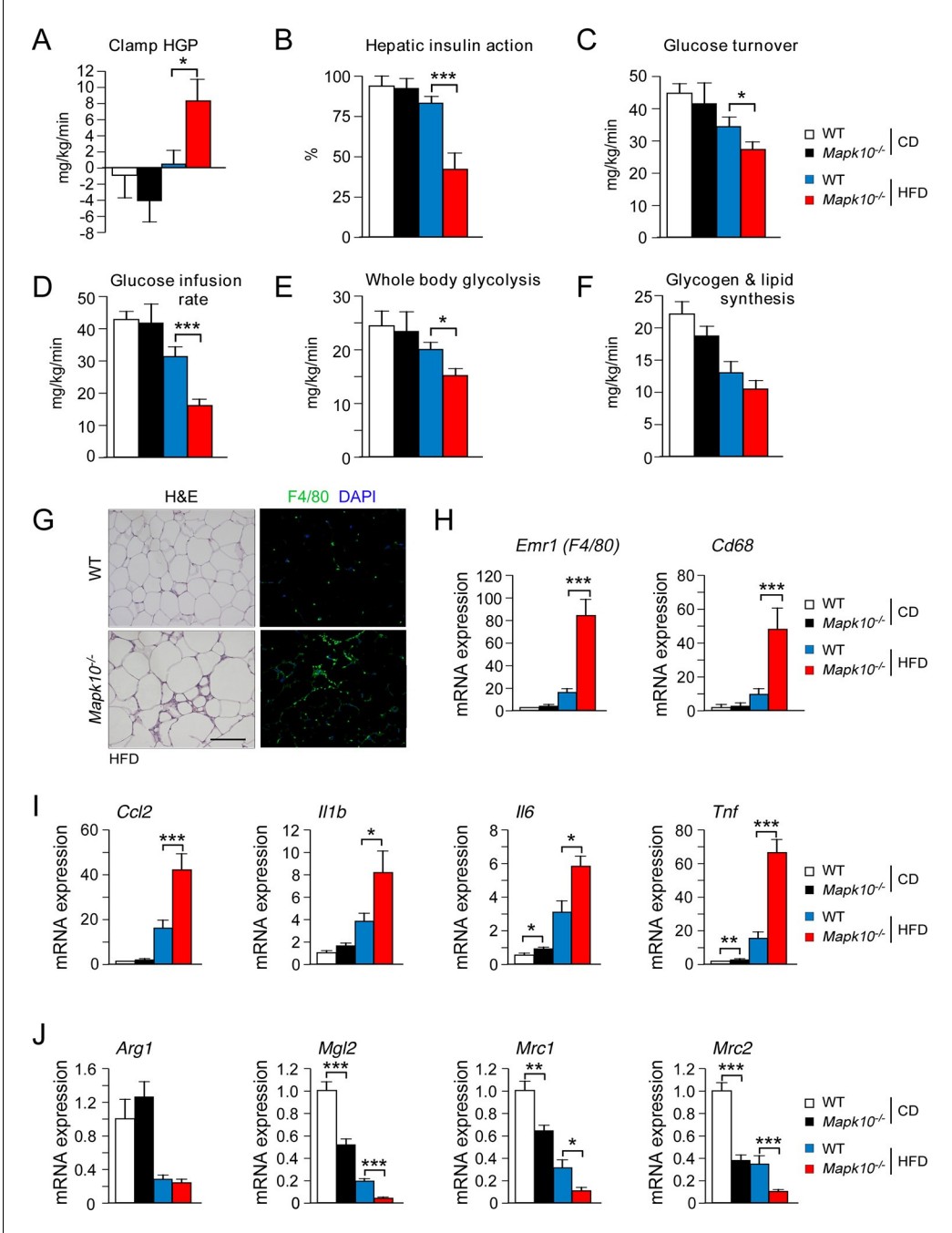

**Figure 2.** JNK3 deficiency promotes and adipose tissue inflammation and insulin resistance. (A-F) Hyperinsulinemic-euglycemic clamps were performed on CD-fed or HFD-fed (3 wk) WT and *Mapk10$^{-/-}$* mice. Clamp hepatic glucose production (A), hepatic insulin action (B), glucose turnover (C), glucose infusion rate (D), whole body glycolysis (E), and glycogen plus lipid synthesis (F) were measured (mean ± SE; n=8; *p<0.05; ***p<0.001). (G-J) Sections of epididymal WAT from HFD-fed (12 wk) WT and *Mapk10$^{-/-}$* mice were stained with hematoxylin & eosin or with an antibody to the macrophage protein F4/80 (G). Macrophage infiltration was examined by measurement of the expression of *Cd68* and *Emr1 (F4/80)* mRNA (H) and also mRNA expressed by genes associated with M1-like (I) and M2-like (J) polarization by Taqman© assays (mean ± SEM; n=10~12; *p<0.05; **p<0.01; ***p<0.001).

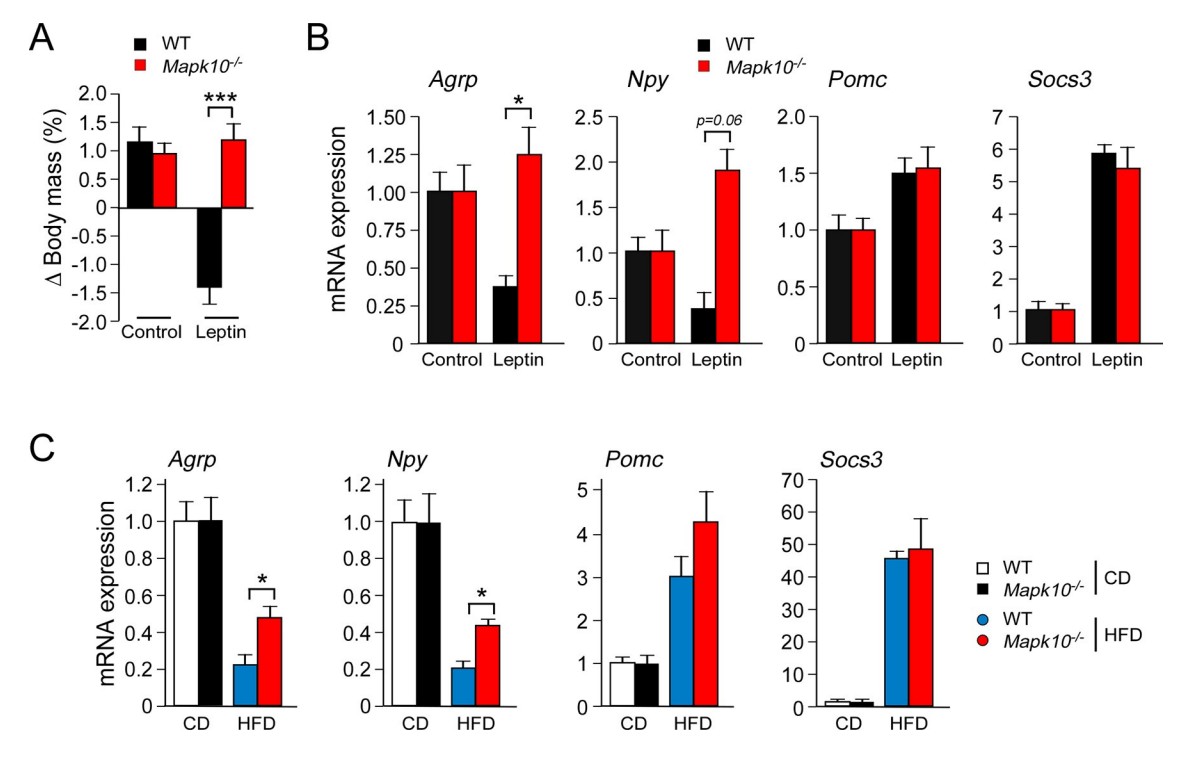

**Figure 3.** JNK3 deficiency causes a selective defect in AgRP neurons. (**A**) HFD-fed (4 wk) WT and *Mapk10⁻ᐟ⁻* mice were treated by intracerebroventricular administration of 5 µg leptin or solvent (Control). The change in body mass at 24 hr post-treatment was measured (mean ± SEM; n=10~12; ***p<0.001). (**B**) WT and *Mapk10⁻ᐟ⁻* mice were treated without or with leptin (2h) prior to measurement of hypothalamic gene expression by Taqman© assays (mean ± SEM; n=10~12; *p<0.05). (**C**) Hypothalamic gene expression in CD-fed and HFD-fed (12 wk) WT and *Mapk10⁻ᐟ⁻* mice was measured by Taqman© assay (mean ± SEM; n=10~12; *p<0.05).

was significantly greater in HFD-fed *Mapk10⁻ᐟ⁻* mice compared with HFD-fed WT mice (*Figure 1G*). These changes in the amount of leptin circulating in the blood correlate, as expected, with differences in obesity (*Friedman, 2014*). However, the hyperleptinemia and hyperphagia of HFD-fed *Mapk10⁻ᐟ⁻* mice is not consistent with the established anorexigenic function of leptin. This analysis suggested that leptin signaling may be suppressed in HFD-fed *Mapk10⁻ᐟ⁻* mice. To test this hypothesis, we examined the effect of treating mice with leptin. We found that intracerebroventricular administration of leptin decreased the body mass of WT mice, but not *Mapk10⁻ᐟ⁻* mice (*Figure 3A*). Measurement of hypothalamic gene expression demonstrated that leptin decreased *Agrp* and *Npy* expression in WT mice, but not *Mapk10⁻ᐟ⁻* mice (*Figure 3B*). In contrast, leptin caused increased *Pomc* and *Socs3* gene expression in both WT and *Mapk10⁻ᐟ⁻* mice (*Figure 3B*). These data indicate that *Mapk10⁻ᐟ⁻* mice exhibit a selective deficiency in leptin regulation of *Agrp* and *Npy* expression. To confirm this conclusion, we compared hypothalamic gene expression in CD-fed and HFD-fed mice. This analysis demonstrated increased *Agrp* and *Npy* expression in HFD-fed *Mapk10⁻ᐟ⁻* mice compared with HFD-fed WT mice (*Figure 3C*). In contrast, no significant difference in *Pomc* and *Socs3* gene expression between HFD-fed WT and *Mapk10⁻ᐟ⁻* mice was detected (*Figure 3C*). These observations indicate that JNK3 deficiency caused a selective defect in leptin signaling.

## LEPRb⁺ neurons mediate the effects of JNK3 on feeding behavior

To examine the mechanism of JNK3 function, we established floxed *Mapk10* mice to investigate the neuron-specific effects of JNK3 on feeding behavior (*Figure 4—figure supplement 1*). We tested whether JNK3 in neurons that express the leptin receptor LEPRb regulates feeding behavior by investigating the effect of *Mapk10* gene ablation specifically in LEPRb⁺ neurons. This analysis demonstrated that control *Leprb-cre* (LepR^WT) mice and *Leprb-cre Mapk10^Loxp/LoxP* (LepR^ΔJ3) mice gained similar body mass when fed a CD. However, HFD-fed LepR^ΔJ3 mice gained significant more

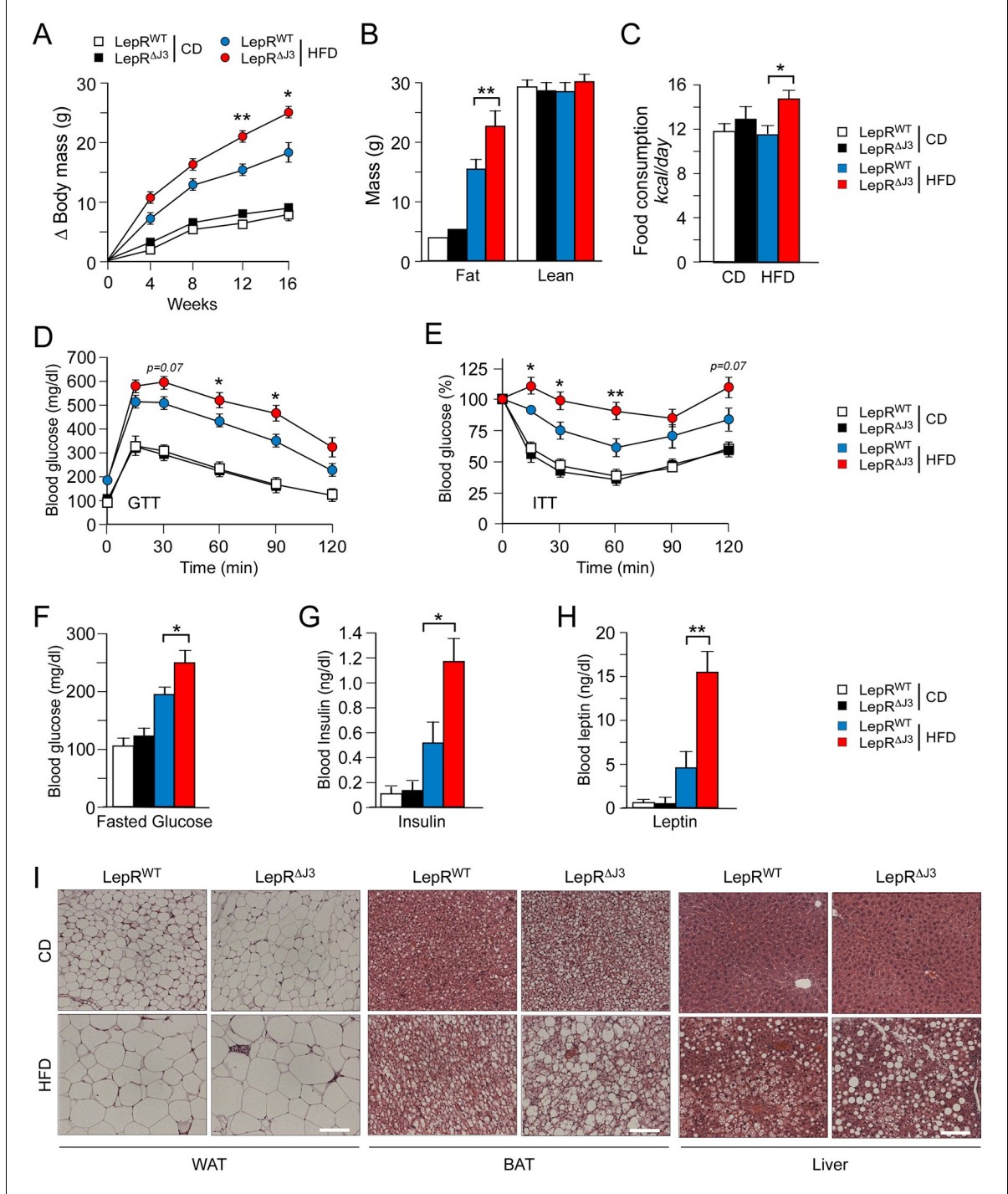

**Figure 4.** JNK3 deficiency in leptin-responsive neurons causes HFD-induced hyperphagia and obesity. (A) The total body mass gain of CD-fed and HFD-fed mice was examined (mean ± SEM; n = 10~25; *p<0.05; **p<0.01). JNK3 deficiency in LEPRb[+] neurons was studied by comparing *Leprb-cre* control mice (LepRb[WT] mice) and *Leprb-cre Mapk10[LoxP/LoxP]* mice (LepR[ΔJ3] mice). (B) The fat and lean mass of CD-fed and HFD-fed (16 wk) mice was measured by [1]H-MRS analysis (mean ± SEM; n = 8~10; **p<0.001). (C) Food consumption by CD-fed and HFD-fed (4 wk) LepR[WT] and LepR[ΔJ3] mice was examined (mean ± SEM; n = 8; *p<0.05). (D,E) Glucose tolerance (D) and insulin tolerance (E) tests were performed using CD-fed and HFD-fed (12 wk) LepR[WT] and LepR[ΔJ3] mice (mean ± SEM; n = 8~12; *p<0.05; **p<0.01). (F-H) CD-fed and HFD-fed (12 wks) LepR[WT] and LepR[ΔJ3] mice were fasted overnight and the blood concentration of glucose (F), insulin (G), and leptin (H) was measured (mean ± SEM; n = 8~20; *p<0.05**p<0.01). (I) Sections of epididymal WAT, interscapular BAT, and liver from CD-fed and HFD-fed (12 wk) LepR[WT] and LepR[ΔJ3] mice were stained with hematoxylin & eosin. Bar, 100 μm.

*Figure 4 continued on next page*

*Figure 4 continued*

The following figure supplements are available for figure 4:

**Figure supplement 1.** Establishment of *Mapk10*[LoxP/LoxP]mice.

**Figure supplement 2.** JNK3 deficiency in leptin-responsive neurons causes obesity.

body mass than LepR[WT] mice (*Figure 4A* and *Figure 4—figure supplement 2A*). [1]H-MRS analysis showed that the difference in body mass was caused by increased fat mass (*Figure 4B*). Metabolic cage analysis demonstrated that *Mapk10* gene ablation in LEPRb[+] neurons caused no change in CD food consumption, but caused increased HFD food consumption (*Figure 4C*). This increase in HFD consumption was not associated with changes in $V_{O2}$, $V_{CO2}$, or energy expenditure (*Figure 4—figure supplement 2B*). JNK3 in LEPRb[+] neurons of HFD-fed mice therefore regulates feeding behavior, but not other aspects of energy balance.

The increased feeding behavior of HFD-fed (but not CD-fed) LepR[ΔJ3] mice was associated with decreased glucose tolerance (*Figure 4D*), increased insulin resistance (*Figure 4E*), increased blood glucose concentration (*Figure 4F*), increased hyperinsulinemia (*Figure 4G*), and increased hyperleptinemia (*Figure 4H*). White and brown adipose tissue (WAT & BAT) in HFD-fed LepR[ΔJ3] mice exhibited increased adipocyte hypertrophy compared with HFD-fed LepR[WT] mice (*Figure 4I*). Moreover, JNK3 deficiency in LepRb[+] neurons caused increased HFD-induced hepatic steatosis (*Figure 4I*).

## JNK3 in AgRP neurons suppresses HFD feeding behavior

To identify a LepRb[+] neuronal sub-population relevant to JNK3-regulated HFD feeding behavior, we examined *Mapk10* gene ablation in selected neurons within the hypothalamus. Gene expression analysis demonstrated that JNK3 was required for HFD-induced regulation of *Agrp* and *Npy*, but not *Pomc* (*Figure 3*). This analysis indicated that AgRP neurons rather than POMC neurons may play an important role in JNK3-regulated feeding behavior in HFD-fed mice. To test this hypothesis, we examined the phenotype of *Agrp-cre Mapk10*[Loxp/LoxP] (Agrp[ΔJ3]) mice and *Pomc-cre Mapk10*[Loxp/LoxP] (Pomc[ΔJ3]) mice. We found that JNK3 deficiency in POMC neurons of HFD-fed mice caused no significant changes in feeding behavior, glucose intolerance, blood glucose concentration, hypertrophy of white and brown adipocytes, and hepatic steatosis compared with control *Pomc-cre* (Pomc[WT]) mice (*Figure 5A,C,E*). In contrast, JNK3 deficiency in AgRP neurons in HFD-fed mice caused increased feeding, increased glucose intolerance, increased blood glucose concentration, increased hypertrophy of white and brown adipocytes, and increased hepatic steatosis compared with control *Agrp-cre* (Agrp[WT]) mice (*Figure 5B,D,F*). Metabolic cage analysis demonstrated that the $V_{O2}$, $V_{CO2}$, and energy expenditure of HFD-fed Agrp[ΔJ3] mice and Pomc[ΔJ3] mice were similar to control mice (*Figure 5—figure supplement 1*). Together, these data demonstrate that JNK3 in AgRP neurons, but not POMC neurons, acts to suppress HFD consumption.

## JNK3 regulates excitatory transmission onto AgRP neurons of HFD-fed mice

Leptin and its receptor are known to affect synaptic transmission and modulate AgRP neuron activity (*Pinto et al., 2004*; *Baver et al., 2014*). We therefore examined miniature inhibitory postsynaptic currents (mIPSCs) and miniature excitatory postsynaptic currents (mEPSCs) of AgRP neurons in the ARC of WT and *Mapk10*[-/-] mice. This analysis demonstrated that JNK3 deficiency caused no change in mIPSC frequency or amplitude in CD-fed and HFD-fed mice (*Figure 6A–D*). Similarly, JNK3 deficiency caused no change in mEPSC frequency or amplitude in CD-fed mice (*Figure 6E–H*). In contrast, HFD-fed JNK3-deficient mice demonstrated increased mEPSC amplitudes in the absence of changes in mEPSC frequency (*Figure 6E–H*). Studies using the selective antagonist DNQX demonstrated that these mEPSC currents were mediated by AMPA receptors in AgRP neurons (*Figure 6-figure supplement 1*). Together, these data indicate that JNK3 deficiency leads to altered excitatory transmission onto AgRP neurons compared to WT mice when fed a HFD. This finding is consistent with the increased expression of AgRP and NPY (*Figure 3C*) and the increased food consumption (*Figure 1F*) observed in HFD-fed JNK3-deficient compared to HFD-fed WT mice.

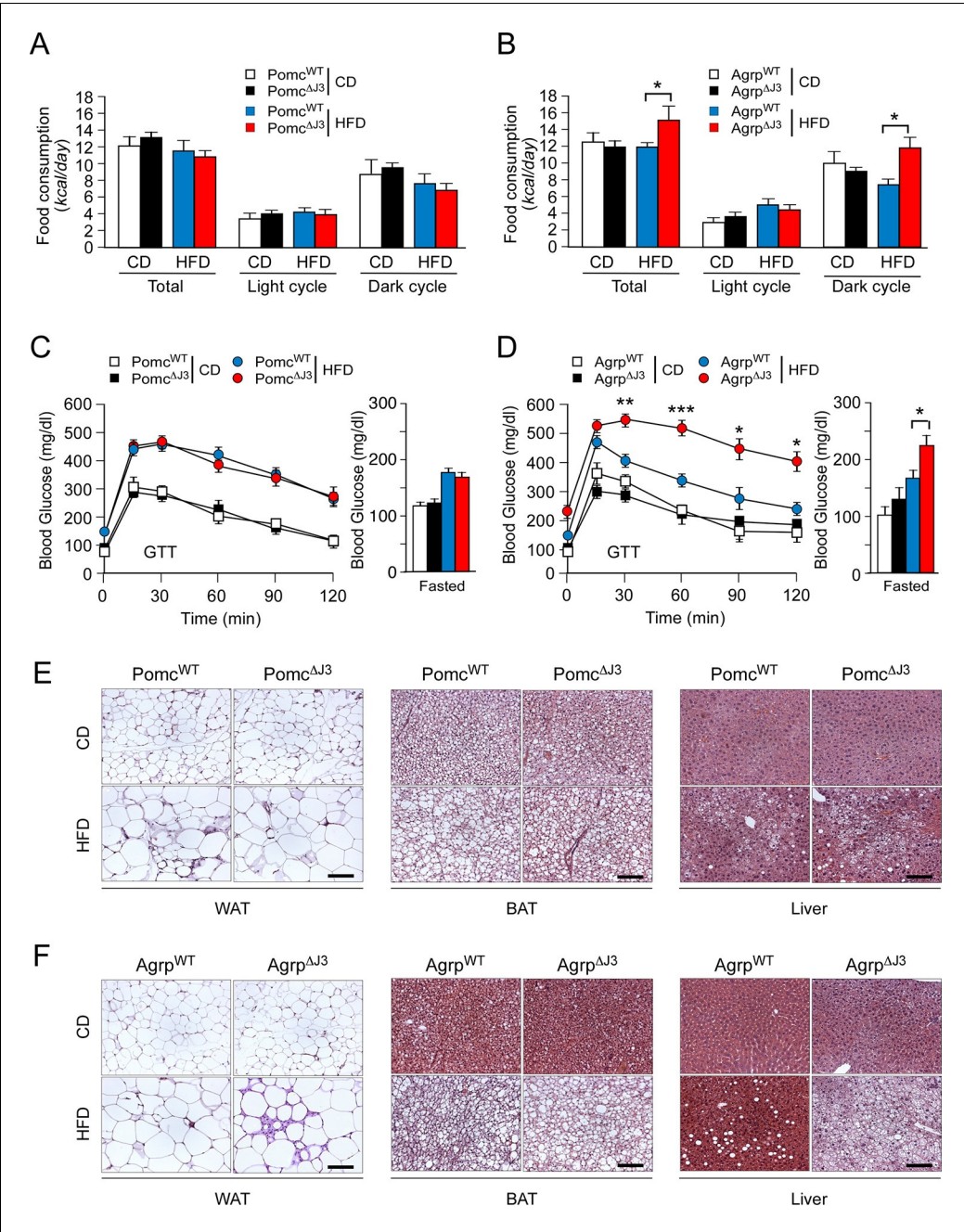

**Figure 5.** JNK3 in AgRP neurons, but not POMC neurons, regulates food consumption. (**A**,**B**) Food consumption by CD-fed and HFD-fed (4 wk) mice was measured (mean ± SEM; n = 8; *p<0.05). JNK3 deficiency in POMC neurons was studied by comparing *Pomc-cre* control mice (Pomc^WT mice) and *Pomc-cre Mapk10^LoxP/LoxP* mice (Pomc^ΔJ3 mice). JNK3 deficiency in AgRP neurons was studied by comparing *Agrp-cre* control mice (Agrp^WT mice) and *Agrp-cre Mapk10^LoxP/LoxP* mice (Agrp^ΔJ3 mice). (**C**,**D**) CD-fed and HFD-fed (16 wk) control mice and mice with JNK3 deficiency in POMC neurons (**C**) and AgRP neurons (**D**) or were tested using glucose tolerance assays and by measurement of fasting blood glucose concentration (mean ± SEM; n = 8~12; *p<0.05; **p<0.01; ***p<0.001). (**E**, **F**) Representative hematoxylin & eosin-stained sections of liver, epididymal WAT, and interscapular BAT from CD-fed and HFD-fed (16 wk) control mice and mice with JNK3 deficiency in POMC neurons (**E**) and AgRP neurons (**F**) are presented.

The following figure supplement is available for figure 5:

**Figure supplement 1.** Effect of JNK3 deficiency in AgRP and POMC neurons on energy expenditure.

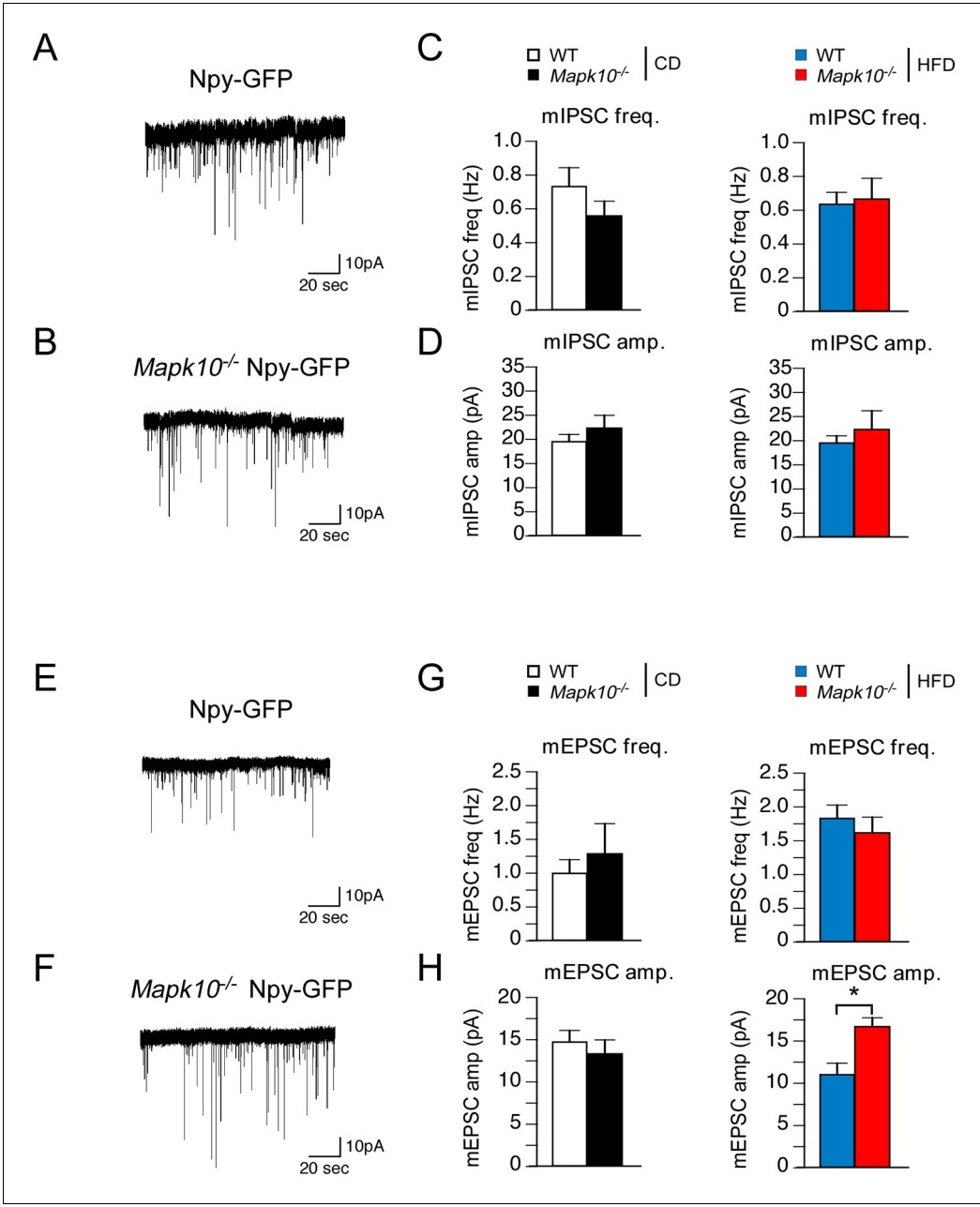

**Figure 6.** JNK3 regulates excitatory transmission onto AgRP neurons. (**A,B**) *Mapk10^{+/+} Npy-GFP* and *Mapk10^{-/-} Npy-GFP* mice were fed a HFD (3 wk) prior to electrophysiological recording of mIPSC from AgRP neurons. (**C,D**) mIPSC frequency (freq.) and amplitude (amp.) in recordings of CD-fed and HFD-fed mice were quantitated (mean ± SEM; n=11~12; *p<0.05). (**E,F**) *Mapk10^{+/+} Npy-GFP* and *Mapk10^{-/-} Npy-GFP* mice were fed a HFD (3 wk) prior to electrophysiological recording of mEPSC from AgRP neurons. (**G,H**) mEPSC frequency and amplitude in recordings of CD-fed and HFD-fed mice were quantitated (mean ± SEM; n=10; *p<0.05).

The following figure supplement is available for figure 6:

**Figure supplement 1.** The AMPA receptor antagonist DNQX blocks mEPSCs in AgRP neurons.

## Discussion

The JNK signaling pathway is implicated in the metabolic stress response (*Sabio and Davis, 2010*). Studies of the ubiquitously expressed isoforms JNK1 and JNK2 demonstrate that the JNK pathway

is activated in peripheral tissues by feeding a HFD (*Hirosumi et al., 2002*). Consequences of HFD-stimulated JNK1 and JNK2 activation in peripheral tissues include promotion of insulin resistance in adipose tissue, liver, and muscle (*Sabio et al., 2008*; *Sabio et al., 2010b*; *Vernia et al., 2014*). In contrast, central actions of JNK1 and JNK2 are mediated by the hypothalamic-pituitary axis by regulation of energy expenditure (*Belgardt et al., 2010*; *Sabio et al., 2010a*; *Vernia et al., 2013*). Together, these studies indicate that JNK1 and JNK2 play important roles in metabolic stress responses by causing insulin resistance in peripheral tissues and promoting obesity by suppressing energy expenditure (*Sabio and Davis, 2010*).

JNK3 is expressed in a limited number of tissues, including the brain and testis (*Gupta et al., 1996*). Since JNK1 and JNK2 are expressed ubiquitously, the expression of JNK3 by neurons means that these cells express all three JNK isoforms (*Davis, 2000*). To examine the role of JNK in neurons, the effects of ablation of the three genes that encode JNK (*Mapk8, Mapk9,* and *Mapk10*) in neurons have been examined. This analysis demonstrated that compound JNK-deficiency caused markedly increased survival responses associated with increased autophagy (*Xu et al., 2011*). Roles for individual JNK isoforms in neurons have also been studied (*Coffey, 2014*). JNK1 and, to some extent JNK2, are constitutively activated and are primarily localized to axons and dendrites (*Coffey et al., 2000*; *Oliva et al., 2006*) where they play a major role in the regulation of the cytoskeleton and axonal/dendritic morphology (*Coffey, 2014*). In contrast, JNK3 exhibits low basal activity and is activated in the nucleus when neurons are exposed to stress (*Yang et al., 1997*). Studies of *Mapk10*[-/-] mice demonstrate that JNK3 is required for stress-induced cJun phosphorylation and AP-1 activation in neurons (*Yang et al., 1997*). This role of JNK3 in neurons is non-redundant with JNK1 and JNK2.

Here we report that JNK3 in LEPRb[+] neurons regulates feeding behavior in mice (*Figure 4*). The mechanism of JNK3 function requires metabolic stress (e.g. feeding a HFD) to cause JNK3 activation. This distinguishes the JNK3 deficiency phenotype from other negative regulators of leptin signaling. Thus, JNK3 deficiency does not cause hyperphagia when mice are fed a chow diet, but JNK3 deficiency does cause hyperphagia when mice are fed a HFD. In contrast, PTPN1-deficiency causes hypophagia on both CD and HFD (*Bence et al., 2006*). This analysis indicates that JNK3 is not required for fine-tuning leptin receptor signaling, but JNK3 is essential for determining the leptin signaling response during exposure to metabolic stress. JNK3 therefore serves a key role in the establishment of the set-point for the threshold of leptin signaling that controls feeding behavior in response to metabolic stress.

Gene ablation studies in sub-populations of LEPRb[+] neurons demonstrated that HFD (but not CD) hyperphagia was found in mice with JNK3 deficiency in AgRP neurons, but not POMC neurons (*Figure 5*). These data demonstrate that JNK3 deficiency in AgRP neurons is sufficient to cause HFD hyperphagia, although possible roles for JNK3 in other LEPRb[+] neurons cannot be excluded by this analysis. We conclude that orexigenic signaling by AgRP neurons contributes to the effects of JNK3 deficiency on HFD hyperphagia.

Molecular mechanisms that account for JNK3 function include altered excitatory transmission to AgRP neurons in HFD-fed mice. Our recordings measured glutamatergic transmission from all inputs to AgRP neurons and demonstrated an increase in mEPSC amplitude, but not frequency, from HFD-fed JNK3-deficient mice compared with HFD-fed control mice (*Figure 6*). This observation is consistent with a possible postsynaptic function of JNK3 in AgRP neurons whereby JNK3 affects AMPA and/or NMDA receptor activity within these neurons. Interestingly, glutamatergic input to AgRP neurons stimulates feeding behavior (*Liu et al., 2012*). Previous studies have established functional connections between the JNK signaling pathway and glutamatergic receptor signaling in neurons. For example, the JNK scaffold proteins JIP1/2 can regulate NMDA receptor signaling (*Kennedy et al., 2007*) and AMPA receptor phosphorylation by JNK regulates AMPA receptor function and trafficking (*Thomas et al., 2008*). Further studies are required to identify the complete spectrum of JNK3 targets in AgRP neurons. Nevertheless, since an increased AMPA response was detected in JNK3-deficient AgRP neurons (*Figure 6H* and *Figure 6-figure supplement 1*), we conclude that JNK-mediated AMPA receptor regulation (*Thomas et al., 2008*) may contribute to the hyperphagic phenotype of HFD-fed JNK3-deficient mice.

The results of the present study indicate that JNK3 plays a major role in the regulation of energy balance. This function of JNK3 to regulate feeding behavior differs from the roles of JNK1/JNK2 to regulate energy expenditure and insulin resistance (*Sabio and Davis, 2010*). These conclusions are based on loss-of-function studies. A contrasting conclusion has been reported based on gain-of-

function studies using transgenic expression of a MKK7-JNK1 fusion protein (that mimics constitutively activated JNK1) in AgRP neurons that causes a small increase in food consumption by CD-fed mice (*Tsaousidou et al., 2014*). Since JNK1-deficient (*Mapk8$^{-/-}$*) mice do not exhibit altered feeding behavior (*Sabio et al., 2008*) and endogenous JNK1 is constitutively activated in neurons (*Coffey, 2014*), it is unclear why transgenic over-expression of an activated *Mapk8* allele (encoding a MKK7-JNK1 fusion protein) in WT mice would cause a small change in feeding behavior. However, the pro-apoptotic function of this activated *Mapk8* allele (*Lei et al., 2002*) may cause defects in hypothalamic neuronal circuits that contribute to the reported phenotype. On balance, we favor the conclusion that JNK1 and JNK2 do not influence feeding behavior (*Sabio and Davis, 2010*), but JNK3 promotes leptin-mediated suppression of HFD feeding behavior (when JNK3 is activated), but not CD feeding behavior (when JNK3 is inactive).

The observation that JNK1 and JNK2 promote obesity (by inhibiting energy expenditure) and cause insulin resistance in peripheral tissues indicates that drugs that block JNK signaling may be therapeutically beneficial for the treatment of pre-diabetes (*Sabio and Davis, 2010*). However, this study demonstrates that JNK3 inhibition causes HFD-dependent hyperphagia (*Figure 1F*). This represents a potential problem for drug therapy. While JNK1/2 inhibition may be therapeutically beneficial, hyperphagia may therefore result from JNK3 inhibition. Consequently, the most effective drug strategy for the treatment of pre-diabetes may require a small molecule that inhibits JNK1/2, but not JNK3.

## Materials and methods

### Mice

We have described *Mapk10$^{-/-}$* mice previously (*Yang et al., 1997*). We obtained C57BL/6J mice (stock number 000664), B6.129S4-*Gt(ROSA)26Sor$^{tm1(FLP1)Dym}$*/RainJ (*Farley et al., 2000*) (stock number 009086), B6.129-*Lepr$^{tm2(cre)Rck}$*/J mice (*DeFalco et al., 2001*) (stock number 008320), B6.FVB-Tg (Npy-hrGFP)1Lowl/J mice (*van den Pol et al., 2009*) (stock number 006417), *Agrp$^{tm1(cre)Lowl}$*/J mice (*Tong et al., 2008*) (stock number 012899), and Tg(Pomc1-cre)16Lowl/J mice (*Balthasar et al., 2004*) (stock number 005965) from the Jackson Laboratory. These mice were backcrossed to the C57BL/6J genetic background.

We established *Mapk10$^{LoxP/LoxP}$* mice using homologous recombination in C57BL/6N embryonic stem cells, the generation of chimeric mice, and breeding to obtain germ-line transmission of the floxed *Mapk10* allele using standard procedures. The mice used for these studies were backcrossed to the C57BL/6J strain. The *Frt-Neo* cassette was excised by crossing the mice with FLP transgenic mice. Homologous recombination of 5′ arm of the targeting vector was verified by PCR using the primers 1F: 5′-TGTGACCTTCTAATACAG-3′ and 2R: 5′-CCTAAGACTGTCAGAGAG-3′ (*Mapk10$^+$*: 135 bp; *Mapk10$^{LoxP}$*: 282 bp). Homologous recombination of the 3′ arm of the targeting vector was verified by PCR using the primers (3F: 5′-CTGAGTGACGTGTGGAG-3′ and 5R: 5′-TCATTGGG TTGGGATATTC-3′) followed by digestion with *XhoI* (*Mapk10$^+$*: 1,975 bp; *Mapk10$^{LoxP}$*: 1026 bp & 1028 bp). *Cre*-mediated recombination between the *LoxP* sites was detected by PCR using the primers 1F and 4R: 5′-GATTCTCCCTGTCTGAG-3′ (*Mapk10$^+$*: 1008 bp; *Mapk10$^{LoxP}$*: 1759 bp; *Mapk10$^Δ$*: 171 bp). The *Mapk10$^{LoxP/LoxP}$* mice were routinely genotyped by PCR using primers 1F and 2R (*Mapk10$^+$*: 135 bp; *Mapk10$^{LoxP}$*: 282 bp).

Male mice (8 wks old) were fed a chow diet (Iso Pro 3000, Purina) or a HFD (F3282, Bioserve) for 4 to 12 wks. Body weight was measured on a weekly basis and whole body fat and lean mass were non-invasively measured using $^1$H-MRS (Echo Medical Systems, Houston, TX). The mice were housed in a facility accredited by the American Association for Laboratory Animal Care (AALAC). The Institutional Animal Care and Use Committee (IACUC) of the University of Massachusetts and the University of Cincinnati approved all studies using animals.

### Hyperinsulinemic-euglycemic clamp studies

The clamp studies were performed at the National Mouse Metabolic Phenotyping Center at the University of Massachusetts Medical School. A 2 hr hyperinsulinemic-euglycemic clamp was conducted using overnight fasted conscious mice with a primed and continuous infusion of human insulin (150

mU/kg body weight priming followed by 2.5 mU/kg/min; Humulin; Eli Lilly), and 20% glucose was infused at variable rates to maintain euglycemia (*Kim et al., 2004*).

## Metabolic cages

The analysis was performed by the National Mouse Metabolic Phenotyping Centers at the University of Massachusetts Medical School and the University of Cincinnati. The mice were housed under controlled temperature and lighting with free access to food and water. The food/water intake, energy expenditure, respiratory exchange ratio, and physical activity were measured using metabolic cages (TSE Systems, Chesterfield, MO).

## Leptin treatment

Intracerebroventricular treatment with leptin was performed using mice with a cannula stereotaxically implanted into the 3$^{rd}$ ventricle (coordinates from Bregma: anteroventral, -1.8 mm; lateral, 0.0 mm; dorsoventral, 5.0 mm). Mice were monitored daily and allowed to recover for 1 week after surgery. Mice received either solvent (artificial cerebrospinal fluid; aCSF) or Leptin (5 µg) in 2 µl delivered over 10 min. Leptin treatment by intraperitoneal (ip) injection was performed following 3 consecutive days of sham injection.

## RNA analysis

Tissue isolated from mice starved overnight was used to isolate total RNA using the RNAeasy mini kit (Qiagen). Total RNA (500 ng) was converted into cDNA using the high capacity cDNA reverse transcription kit (Life Technologies, Carlsbad, CA). The diluted cDNA was used for real-time quantitative PCR analysis using a Quantstudio PCR PCR machine (Life Technologies). TaqMan assays (Life Technologies) were used to quantify *Adipoq* (Mm00456425_m1), *Agrp* (Mm00475829_g1), *Arg1* (Mm00475988_m1), *Ccl2* (Mm00441242_m1), *Emr1* (F4/80) (Mm00802530_m1), *Il1b* (Mm00434228_m1), *Il6* (Mm00446190_m1), *Mapk8 (Jnk1)* (Mm00489514_m1), *Mapk9 (Jnk2)* (Mm00444231_m1), *Mapk10 (Jnk3)* (Mm00436518_m1), *Mgl2* (Mm00460844_m1), *Mrc1* (Mm00485148_m1), *Mrc2* (Mm00485184_m1), *Npy* (Mm03048253_m1), *Pomc* (Mm00435874_m1), and *Tnf* (Mm00443258_m1). The relative mRNA expression was normalized by measurement of the amount of *18S* RNA in each sample using Taqman$^{©}$ assays (catalog number 4308329; Life Technologies).

## Blood analysis

Blood glucose was measured with an Ascensia Breeze 2 glucometer (Bayer, Pittsburgh, PA). Adipokines and insulin in plasma were measured by multiplexed ELISA using a Luminex 200 machine (Millipore, Billerica, MA).

## Glucose and insulin tolerance tests

Glucose and insulin tolerance tests were performed by intraperitoneal injection of mice with glucose (1 g/kg) or insulin (1.5 U/kg) using methods described previously (*Sabio et al., 2008*).

## JNK3 activation

Mice (8–12 week-old) were fasted overnight. Hypothalamic extracts were prepared using Triton lysis buffer (20 mM Tris-pH 7.4, 1% Triton-X100, 10% glycerol, 137 mM NaCl, 2 mM EDTA, 25 mM β-glycerophosphate, 1 µM sodium orthovanadate, 1 µM PMSF and 10 µg/mL leupeptin and aprotinin). Extracts (30–50 µg of protein) were examined by immunoblot analysis by probing with antibodies to JNK3 (Cell Signaling Technologies, Danvers, MA) and GAPDH (Santa Cruz Biotechnology, Dallas, TX). Activated JNK was isolated by immunoprecipitation with the mouse monoclonal p-JNK antibody G9 (Cell Signaling Technologies) pre-bound to protein G Sepharose (GE Healthcare, Pittsburgh, PA) and detected by immunoblot analysis by probing with an antibody to JNK3 (Cell Signaling Technologies). Immunocomplexes were detected by fluorescence using anti-mouse and anti-rabbit secondary IRDye antibodies (LI-COR Biosciences, Lincoln, NE) and quantitated using the Li-COR Imaging system

## Analysis of tissue sections

Histology was performed using tissue fixed in 10% formalin for 24 h, dehydrated, and embedded in paraffin. Sections (7 µm) were cut and stained using hematoxylin & eosin (American Master Tech Scientific, Lodi, CA). Paraffin sections were stained with an antibody to F4/80 (Abcam, Cambridge, MA) that was detected by incubation with anti-rabbit Ig conjugated to Alexa Fluor 488 (Life Technologies). DNA was detected by staining with DAPI (Life Technologies). Fluorescence was visualized using a Leica TCS SP2 confocal microscope equipped with a 405 nm diode laser (Leica Microsystems, Buffalo Grove, IL).

## Electrophysiology

Brain slice preparations were performed using 8–10-weeks-old mice anaesthetized with isoflurane before decapitation and removal of the entire brain. The brains were immediately submerged in ice-cold, carbogen-saturated (95% $O_2$, 5% $CO_2$) high sucrose solution (238 mM sucrose, 26 mM $NaHCO_3$, 2.5 mM KCl, 1.0 mM $NaH_2PO_4$, 5.0 mM $MgCl_2$, 10.0 mM $CaCl_2$, 11 mM glucose). Then, 300 µm thick coronal sections were cut with a Leica VT1000S Vibratome and incubated in oxygenated aCSF (126 NaCl, 21.4 mM $NaHCO_3$, 2.5 mM KCl, 1.2 mM $NaH_2PO_4$, 1.2 mM $MgCl_2$, 2.4 mM $CaCl_2$, 10 mM glucose) at 34°C for 30 min. The slices were maintained and recorded at room temperature (20–24°C). The intracellular solution for voltage clamp recording contained the following: 140 mM CsCl, 1 mM BAPTA, 10 mM HEPES, 5 mM $MgCl_2$, 5 mM Mg-ATP, and 0.3 mM $Na_2GTP$, pH 7.35 and 290 mOsm.

To isolate glutamatergic, action potential-independent events, minitature excitatory postsynaptic currents (mEPSCs) were recorded in the presence of tetrodotoxin (1 µM) and picrotoxin (100 µM) in whole cell voltage clamp mode. To record miniature inhibitory postsynaptic currents (mIPSCs), the neurons were recorded in the presence of TTX and kynurenic acid (1 mM). The membrane potential was clamped at −60 mV. All recordings were made using a Multiclamp 700B amplifier, and data were filtered at 1.4 kHz and digitized at 20 kHz. Data was analyzed using Clampfit 10.2 and Origin Pro 8.6.

## Statistical analysis

Differences between groups were examined for statistical significance using the Student's test or analysis of variance (ANOVA) with the Fisher's test.

## Acknowledgements

We thank Dr. David Garlick for pathological analysis of tissue sections and Armanda Roy for technical assistance, and Kathy Gemme for administrative assistance. These studies were supported by grants R01 DK107220 (to RJD) and U24 DK093000 (to JKK) from the National Institutes of Health. RJD and RAF are investigators of the Howard Hughes Medical Institute.

## Additional information

### Competing interests

RJD: Reviewing Editor, *eLife.* The other authors declare that no competing interests exist.

### Funding

| Funder | Grant reference number | Author |
| --- | --- | --- |
| National Institute of Diabetes and Digestive and Kidney Diseases | R01DK107220 | Roger J Davis |
| Howard Hughes Medical Institute | Investigatorship | Richard A Flavell Roger J Davis |
| National Institute of Diabetes and Digestive and Kidney Diseases | U24DK093000 | Jason K Kim |

The funders had no role in study design, data collection and interpretation, or the decision to submit the work for publication.

### Author contributions
SV, CM, JCM, JCK, TB, KC, NJK, DYJ, Final approval of version to be published, Conception and design, Acquisition of data, Analysis and interpretation of data, Drafting or revising the article; JKK, NA, RAF, BBL, RJD, Final approval of version to be published, Conception and design, Analysis and interpretation of data, Drafting or revising the article

### Author ORCIDs
Richard A Flavell, http://orcid.org/0000-0003-4461-0778
Roger J Davis, http://orcid.org/0000-0002-0130-1652

### Ethics
Animal experimentation: This study was performed in strict accordance with the recommendations in the Guide for the Care and Use of Laboratory Animals of the National Institutes of Health. All of the animals were handled according to approved institutional animal care and use committee (IACUC) protocols (#A-978 and #A-1032) of the University of Massachusetts Medical School.

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
