## [Decision Letter]

Thank you for submitting your work entitled "JNK3 participates in an AgRP/NPY neuron circuit that suppresses feeding behavior in response to metabolic stress" for peer review at *eLife*. Your submission has been favorably evaluated by Tony Hunter (Senior Editor) and three reviewers, one of whom is a member of our Board of Reviewing Editors.

The reviewers have discussed the reviews with one another and the Reviewing Editor has drafted this decision to help you prepare a revised submission.

Summary:

All the reviewers agree that this manuscript does a fine job of demonstrating that JNK3 function is important for body weight regulation and that its function in AgRP-expressing neurons is particularly important. However, the mechanism(s) by which JNK3 acts within AgRP neurons is not sufficiently developed. In particular the deficiency in Lepr signaling seems unlikely.

Essential revisions:

For publication in *eLife*, the authors need to delineate the mechanism by which loss of JNK3 signaling in AgRP neurons promotes obesity. The reviewers have suggested several approaches to address this issue. Alternatively, the authors may decide that publishing the more descriptive part of the study (Figure 1–Figure 5) in another journal is the best option.

*Reviewer #1:*

This paper provides evidence that lack of JNK3 expression in all cells, all neurons that express the leptin-receptor or just the sub-population of neurons that expresses AgRP leads to obesity when mice are reared on a high-fat diet (HFD) but not on a normal (control diet). The obesity is associated with elevated leptin, elevated insulin, glucose intolerance, insulin resistance, and many other parameters associated with obesity. The most important contribution is their systematic demonstration that the phenotype of global KO of the gene encoding JNK3 is replicated quite well by KO only in AgRP neurons. The authors go on to explore how loss of JNK3 function in AgRP neurons might lead to the hyperphagic obese phenotype, but those studies are less compelling.

1) The authors show that leptin receptors can be activated (phosphorylated) in the JNK3 KO but the dose-response is apparently shifted – LepR is fully activated (phosphorylated) with 2.5 mg/kg leptin but only partially activated with 0.1 mg/kg (Figure 7B). They attribute this hypo-sensitivity to a reduced level of Lepr mRNA (about half normal). They show that phospho-cJun (transcription factor) normally binds to an AP1 site in the promoter of Lepr gene and hence the lack of phosphorylation of c-Jun could impair transcription of Lepr gene. There are two problems with this analysis: (a) The Lepr mRNA (rather than protein) measurements were made in hypothalamic samples, not in AgRP neurons where it is relevant. (b) Complete loss of Lepr in AgRP neurons has a relatively small effect on obesity and heterozygous Lepr db/+ mice (with half the normal level of Lepr mRNA) do not become more obese on a HFD than WT mice. Thus, it seems unlikely that reduced levels of LepR account for the obesity observed in this JNK3 KO model.

2) Their data in Figure 6 are more promising. They show that mEPSCs amplitude in AgRP neurons is enhanced in slices, suggestive of enhanced excitatory signaling within AgRP neurons. If this enhanced excitability extends to in vivo situation, then that could drive the hyperphagia and ultimately the obesity. Thus, I suggest that the authors do whatever they can to demonstrate that AgRP neurons are hyperactive in vivo. First they need to demonstrate that the mice are hyperphagic *before* they become obese. The experiment in Figure 4 was performed after there was already a difference in body weight. A pair-feeding experiment would also be useful to demonstrate that the obesity is due to hyperphagia. Looking for markers of enhanced neuronal activity in AgRP neurons (e.g. Fos) might also be revealing. However, the ultimate experiment would be to use in vivo recording or calcium imaging of AgRP neurons – techniques that are available to this group.

*Reviewer #2:*

Vernia and colleagues present an interesting paper in which the conclusion are based on a broad set of experiments and well justified. They show that the loss of the neuronally expressed JNK3 is associated with increased obesity due to increased food consumption. This is brought about by the elimination of JNK3 in leptin receptor expressing neurons. They narrow this down further by doing an AgRP-specific elimination and compare that to a POMC neuronal-specific elimination. They demonstrate hyperphagia to be present in only Agrp-specific nulls for JNK3. They go on to demonstrate that this is in fact due at least in part due to reduced leptin receptor expression on AgRP-neurons. All of this happens exclusively in the context of high fat diet exposure.

These are interesting findings that should be of broad interest to the field. It is surprising to find JNK3 necessary for leptin receptor expression and to see it distinct from the role of JNK1/2 in the periphery where their lack results in improvement in the metabolic phenotype due to reduced inflammatory tone.

There is very little to criticize here. The authors address an important question, find a surprising answer of broad interest and base their conclusion on a broad set of genetic data.

The only area that falls short is a further analysis of the signaling mechanisms in place. What other component of leptin signaling are affected by the lack of JNK3? The downstream analysis could be broader. Where does the cell specificity come in? How does JNK3 deficiency lead to reduce *lepR* expression in AgRP neurons, and it does not lead to reduced *lepR* expression in POMC neurons?

*Reviewer #3:*

In this manuscript Vernia et al. show that JNK3 plays a role in inhibiting high-fat diet-induced weight gain and does so through actions at AgRP/NPY neurons, which may include bolstering leptin actions. Overall, the metabolic studies convincingly show that JNK3 deficiency increases HFD-induced weight gain, hyperphagia, adipose hypertrophy, glucose intolerance and insulin resistance. The logical set of cell-type-selective JNK3 deletions provides compelling evidence that JNK3 actions specifically in AgRP neurons are necessary to reduce HFD-induced weight gain. Thus, the overall conclusion is well supported by the data presented in Figure 1–Figure 5.

The primary weakness in the work is the cursory attempt to provide some mechanistic insight (Figures 6 and 7). The mEPSC recordings add very little to the paper as is. It is important to know that mIPSCs don't also increase and how the deletion affects basal regulation of POMC neurons. In Figure 7, the overall JNK3 knockout was used and thus, it is not possible to determine if leptin receptor expression is also reduced in POMC cells or rather, if maybe POMC cells do not express JNK3 or require JNK3 to bolster leptin sensitivity. In my opinion, these last 2 experiments do not need to be included to support the conclusions made, but if they are included, they need to be done more thoroughly so that they can be better interpreted.

JNK3 is the neuron-specific isoform of JNK and its activity is dynamically regulated rather than constitutive like the other 2 isoforms. Together with the compelling evidence provided here that JNK3 helps reduce high-fat diet-induced weight gain through actions in AgRP neurons, this work adds significant new insight into JNK3 function and the hypothalamic regulation of energy balance in the face of metabolic challenge.

[Editors' note: further revisions were requested prior to acceptance, as described below.]

Thank you for submitting your work entitled "Excitatory transmission onto AgRP neurons is regulated by cJun NH_2_-terminal kinase 3 in response to metabolic stress" for consideration by *eLife*. Your article has been evaluated by a Reviewing Editor and overseen by Tony Hunter as the Senior Editor.

In response to our reviews, the authors of this paper have now dropped data (old Figure 7) suggesting that reduced leptin receptor expression was responsible for the obesity of JNK3-deficient mice in favor enhanced glutamatergic (mEPSC) signaling (Figure 6). The authors have added panels to Figure 6 showing that inhibitory (mIPSC) is unchanged. Overall, the hypothesis that enhanced glutamatergic signaling (excitation) of AgRP neurons would maintain *Npy* and *Agrp* expression and enhance feeding behavior makes sense, especially because the previous work from the Lowell lab (Liu et al., 2012) showing that removing NMDA receptors from AgRP neurons has the opposite effect. However, there are a few loose ends that need attention.

1) Unfortunately, the electrophysiology experiments were performed on *Mapk10* KO mice rather than mice lacking *Mapk10* selectively in AgRP neurons; hence, it is not possible to make a strong conclusion that the enhanced excitatory transmission is due to lack of JNK3 in AgRP neurons.

2) The authors suggest in the Discussion that NMDA signaling is enhanced, but their experiments do not distinguish NMDA from AMPA signaling.

3) The hypothesis that links JNK3 deficiency and the presumptive enhanced NMDA signaling is not clearly explained. In the fifth paragraph of the Discussion the authors say that JIP proteins might be involved and refer to Kennedy et al. (2007). If I understand that paper correctly, it suggests that lack of JIP proteins influences JNK localization (activity) and prevents phosphorylation of NR2 subunit of NMDA receptor. If so, it seems like lack of JNK3 would impair (not enhance) NMDA function in the present studies. The authors admit that "further studies are required to identify the complete spectrum of JNK3 targets in AgRP neurons". I would argue that they have not identified any direct JNK3 targets in AgRP neurons.

Essential revisions:

At the very least, the authors need to discern whether AMPA or NMDA receptor signaling is altered in JNK3-deficient AgRP neurons and provide some data or reasonable hypothesis indicating how lack of JNK3 could lead to enhanced AMPA/NMDA signaling.

---

## [Author Response]

*Essential revisions:*

*For publication in eLife, the authors need to delineate the mechanism by which loss of JNK3 signaling in AgRP neurons promotes obesity. The reviewers have suggested several approaches to address this issue. Alternatively, the authors may decide that publishing the more descriptive part of the study (Figure 1–Figure 5) in another journal is the best option.*

Thank you for providing this opportunity to improve our manuscript. We have carefully considered the reviewers’ comments and we have thoroughly revised the manuscript by including new experiments to more fully establish our conclusions. Specifically, we demonstrate that JNK3 plays no role in the regulation of feeding behavior in chow-fed mice, but activated JNK3 in high fat diet-fed mice regulates excitatory transmission onto ARP/NPY neurons. Collectively, our data establish a new paradigm for the regulation of feeding behavior by metabolic stress.

*Reviewer #1: This paper provides evidence that lack of JNK3 expression in all cells, all neurons that express the leptin-receptor or just the sub-population of neurons that expresses AgRP leads to obesity when mice are reared on a high-fat diet (HFD) but not on a normal (control diet). The obesity is associated with elevated leptin, elevated insulin, glucose intolerance, insulin resistance, and many other parameters associated with obesity. The most important contribution is their systematic demonstration that the phenotype of global KO of the gene encoding JNK3 is replicated quite well by KO only in AgRP neurons. The authors go on to explore how loss of JNK3 function in AgRP neurons might lead to the hyperphagic obese phenotype, but those studies are less compelling. 1) The authors show that leptin receptors can be activated (phosphorylated) in the JNK3 KO but the dose-response is apparently shifted – LepR is fully activated (phosphorylated) with 2.5 mg/kg leptin but only partially activated with 0.1 mg/kg (Figure 7B). They attribute this hypo-sensitivity to a reduced level of Lepr mRNA (about half normal). They show that phospho-cJun (transcription factor) normally binds to an AP1 site in the promoter of Lepr gene and hence the lack of phosphorylation of c-Jun could impair transcription of Lepr gene. There are two problems with this analysis: (a) The Lepr mRNA (rather than protein) measurements were made in hypothalamic samples, not in AgRP neurons where it is relevant. (b) Complete loss of Lepr in AgRP neurons has a relatively small effect on obesity and heterozygous Lepr db/+ mice (with half the normal level of Lepr mRNA) do not become more obese on a HFD than WT mice. Thus, it seems unlikely that reduced levels of LepR account for the obesity observed in this JNK3 KO model.*

We agree. These data have been removed from the revised manuscript. Conclusions about leptin receptor expression have also been removed from the text. The mechanistic analysis in the revised manuscript focuses on the role of JNK3 to regulate excitatory transmission onto ARP/NPY neurons.

*2) Their data in Figure 6 are more promising. They show that mEPSCs amplitude in AgRP neurons is enhanced in slices, suggestive of enhanced excitatory signaling within AgRP neurons. If this enhanced excitability extends to in vivo situation, then that could drive the hyperphagia and ultimately the obesity. Thus, I suggest that the authors do whatever they can to demonstrate that AgRP neurons are hyperactive in vivo. First they need to demonstrate that the mice are hyperphagic before they become obese. The experiment in Figure 4 was performed after there was already a difference in body weight. A pair-feeding experiment would also be useful to demonstrate that the obesity is due to hyperphagia. Looking for markers of enhanced neuronal activity in AgRP neurons (e.g. Fos) might also be revealing. However, the ultimate experiment would be to use in vivo recording or calcium imaging of AgRP neurons – techniques that are available to this group.*

We have revised the manuscript to address these important points:

A) We performed a time course analysis using metabolic cages to examine when hyperphagia is detected in mice fed a high fat diet. This analysis demonstrates that hyperphagia is detected within 2 days of high fat diet consumption (Figure 1—figure supplement 2). ^1^H-MRS analysis on day 3 demonstrated no obesity phenotype (Figure 1—figure supplement 2). These data demonstrate that the high fat diet hyperphagia phenotype occurs prior to the development of obesity.

B) We performed a pair-feeding study using high fat diet-fed WT and JNK3-deficient mice. This analysis demonstrates that no obesity phenotype was detected when the WT and JNK3-deficient mice consumed the same amount of food (Figure 1—figure supplement 3). These data demonstrate that the greater obesity of high fat diet-fed JNK3-deficient mice compared with WT mice is caused by increased food consumption.

C) We have not performed in vivo recording and calcium imaging of AgRP neurons, as requested; we believe that such studies are beyond the scope of the present manuscript. We did examine cFos staining, but no differences in the number of cFos-positive AgRP neurons were detected between WT and JNK3 KO mice. We have expanded the electrophysiological analysis of AgRP neurons from chow diet-fed mice and high fat diet-fed mice in the revised manuscript (Figure 6). This analysis demonstrates that JNK3-deficiency causes no phenotype in chow diet-fed mice, but does cause increased mEPSC amplitude in high fat diet-fed mice (Figure 6).

*Reviewer #2: […] There is very little to criticize here. The authors address an important question, find a surprising answer of broad interest and base their conclusion on a broad set of genetic data. The only area that falls short is a further analysis of the signaling mechanisms in place. What other component of leptin signaling are affected by the lack of JNK3? The downstream analysis could be broader. Where does the cell specificity come in? How does JNK3 deficiency lead to reduce lepR expression in AgRP neurons, and it does not lead to reduced lepR expression in POMC neurons?*

Based on the comments of the other reviewers, we have deleted the section of the original manuscript on LepRb expression. In the revised manuscript, we have focused the data presentation and the central conclusions of our study on the finding that excitatory transmission onto AgRP neurons is regulated by the JNK3 signaling pathway in HFD-fed mice. This observation provides a mechanism for the HFD-specific hyperphagia of JNK3 KO mice. Further studies will be required to identify direct molecular targets of JNK3 signaling; this analysis is beyond the scope of the present study.

*Reviewer #3: […] The primary weakness in the work is the cursory attempt to provide some mechanistic insight (Figures 6 and 7). The mEPSC recordings add very little to the paper as is. It is important to know that mIPSCs don't also increase and how the deletion affects basal regulation of POMC neurons. In Figure 7, the overall JNK3 knockout was used and thus, it is not possible to determine if leptin receptor expression is also reduced in POMC cells or rather, if maybe POMC cells do not express JNK3 or require JNK3 to bolster leptin sensitivity. In my opinion, these last 2 experiments do not need to be included to support the conclusions made, but if they are included, they need to be done more thoroughly so that they can be better interpreted.*

We agree that the leptin receptor expression data do not strengthen our study. Based on this criticism and the comments of reviewer #1, we have deleted these data from the revised manuscript. We have revised the manuscript to include new data to document mIPSCs (Figure 6) and we have more fully documented the changes in mEPSCs. We believe that these data do strengthen the study.

JNK3 is the neuron-specific isoform of JNK and its activity is dynamically regulated rather than constitutive like the other 2 isoforms. Together with the compelling evidence provided here that JNK3 helps reduce high-fat diet-induced weight gain through actions in AgRP neurons, this work adds significant new insight into JNK3 function and the hypothalamic regulation of energy balance in the face of metabolic challenge.

Thank you for these comments.

[Editors' note: further revisions were requested prior to acceptance, as described below.]

*In response to our reviews, the authors of this paper have now dropped data (old Figure 7) suggesting that reduced leptin receptor expression was responsible for the obesity of JNK3-deficient mice in favor enhanced glutamatergic (mEPSC) signaling (Figure 6). The authors have added panels to Figure 6 showing that inhibitory (mIPSC) is unchanged. Overall, the hypothesis that enhanced glutamatergic signaling (excitation) of AgRP neurons would maintain Npy and Agrp expression and enhance feeding behavior makes sense, especially because the previous work from the Lowell lab (Liu et al., 2012) showing that removing NMDA receptors from AgRP neurons has the opposite effect. However, there are a few loose ends that need attention.*

*1) Unfortunately, the electrophysiology experiments were performed on Mapk10 KO mice rather than mice lacking Mapk10 selectively in AgRP neurons; hence, it is not possible to make a strong conclusion that the enhanced excitatory transmission is due to lack of JNK3 in AgRP neurons.*

*2) The authors suggest in the Discussion that NMDA signaling is enhanced, but their experiments do not distinguish NMDA from AMPA signaling.*

*3) The hypothesis that links JNK3 deficiency and the presumptive enhanced NMDA signaling is not clearly explained. In the fifth paragraph of the Discussion the authors say that JIP proteins might be involved and refer to Kennedy et al. (2007). If I understand that paper correctly, it suggests that lack of JIP proteins influences JNK localization (activity) and prevents phosphorylation of NR2 subunit of NMDA receptor. If so, it seems like lack of JNK3 would impair (not enhance) NMDA function in the present studies. The authors admit that "further studies are required to identify the complete spectrum of JNK3 targets in AgRP neurons". I would argue that they have not identified any direct JNK3 targets in AgRP neurons.*

Our focus on JNK3 function in AgRP neurons is based on the findings that:

1) JNK3-deficiency causes changes in *Agrp* and *Npy* expression, but not *Pomc* expression, in the response to leptin administration (Figure 3) or feeding a HFD (Figure 3);

2) mEPSC amplitudes are increased in AgRP neurons of JNK3-deficient mice (Figure 6); and

3) JNK3-deficiency in AgRP neurons is sufficient to cause hyperphagy on a HFD (Figure 5).

Together, these data provide strong evidence for a function of JNK3 in AgRP neurons. We agree that electrophysiological studies of *AgRP-cre^+^ Mapk10^LoxP/LoxP^ NPY-GFP* mice would be interesting, but these mice are not available in our colony; we consider these studies to be beyond the scope of the present study.

We apologize for the poor presentation of the conclusions of our study in the Discussion section of our manuscript where we suggested that NMDA receptors may be regulated (based on the two papers we cited) without adequately addressing the role of AMPA receptors. We have revised the manuscript to make our presentation clearer. Moreover, we have included new data in the manuscript (Figure 6—figure supplement 1) that demonstrates the role of AMPA responses in the mEPSC recordings. These new data have enabled greater focus of the manuscript. *These data represent a key finding that provides a mechanism for JNK3 function mediated by the regulation of AMPA responses*. It is established that AMPA receptor phosphorylation by JNK controls AMPA receptor trafficking and function (Thomas et al. 2008).

Although an altered AMPA response in AgRP neurons contributes to the JNK3 phenotype, we cannot exclude the possibility that there is also an altered NMDA response. We feel that dissecting the specific contributions of AMPA and NMDA receptors, while interesting, is outside the scope of this study. In addition, while our recordings show alterations in mEPSC amplitudes, further studies are needed to identify the full spectrum of JNK3 targets that could potentially result in this observation. Such targets may include the JNK scaffold protein JIP1 (that inhibits NMDAR signaling) and JIP2 (that increases NMDAR signaling), but we consider studies of these scaffold proteins to be beyond the scope of the present manuscript.

Essential revisions:

*At the very least, the authors need to discern whether AMPA or NMDA receptor signaling is altered in JNK3-deficient AgRP neurons and provide some data or reasonable hypothesis indicating how lack of JNK3 could lead to enhanced AMPA/NMDA signaling.*

We agree – this is an important point. We did not make conclusions about AMPA vs. NMDA currents in the previous version of this manuscript. We have therefore revised the manuscript to include new data documenting that the mEPSC currents that we detect are mediated by AMPA receptor responses, not NMDA receptor responses (Figure 6—figure supplement 1). These data have enabled us to focus on AMPA responses and the established role of JNK phosphorylation to regulate AMPA receptor trafficking and function (subsection “JNK3 regulates excitatory transmission onto AgRP neurons of HFD-fed mice“). We believe that our revised manuscript has been substantially improved by incorporation of these changes.